# Applying machine learning in motor activity time series of depressed bipolar and unipolar patients compared to healthy controls

Petter Jakobsen[1,2]*, Enrique Garcia-Ceja[3], Michael Riegler[4,5], Lena Antonsen Stabell[1,2], Tine Nordgreen[6,7], Jim Torresen[5], Ole Bernt Fasmer[1,2], Ketil Joachim Oedegaard[1,2]

**1** NORMENT, Division of Psychiatry, Haukeland University Hospital, Bergen, Norway, **2** Department of Clinical Medicine, University of Bergen, Bergen, Norway, **3** SINTEF Digital, Oslo, Norway, **4** Simula Metropolitan Center for Digitalisation, Oslo, Norway, **5** Department of Informatics, University of Oslo, Oslo, Norway, **6** Division of Psychiatry, Haukeland University Hospital, Bergen, Norway, **7** Department of Clinical Psychology, Faculty of Psychology, University of Bergen, Bergen, Norway

* petter.jakobsen@helse-bergen.no

**Data Availability Statement:** The data can be accessed via: http://datasets.simula.no/depresjon/

## Abstract

Current practice of assessing mood episodes in affective disorders largely depends on subjective observations combined with semi-structured clinical rating scales. Motor activity is an objective observation of the inner physiological state expressed in behavior patterns. Alterations of motor activity are essential features of bipolar and unipolar depression. The aim was to investigate if objective measures of motor activity can aid existing diagnostic practice, by applying machine-learning techniques to analyze activity patterns in depressed patients and healthy controls. Random Forrest, Deep Neural Network and Convolutional Neural Network algorithms were used to analyze 14 days of actigraph recorded motor activity from 23 depressed patients and 32 healthy controls. Statistical features analyzed in the dataset were mean activity, standard deviation of mean activity and proportion of zero activity. Various techniques to handle data imbalance were applied, and to ensure generalizability and avoid overfitting a Leave-One-User-Out validation strategy was utilized. All outcomes reports as measures of accuracy for binary tests. A Deep Neural Network combined with SMOTE class balancing technique performed a cut above the rest with a true positive rate of 0.82 (sensitivity) and a true negative rate of 0.84 (specificity). Accuracy was 0.84 and the Matthews Correlation Coefficient 0.65. Misclassifications appear related to data overlapping among the classes, so an appropriate future approach will be to compare mood states intra-individualistically. In summary, machine-learning techniques present promising abilities in discriminating between depressed patients and healthy controls in motor activity time series.

Or directly downloaded from: https://doi.org/10.5281/zenodo.1219550.

**Funding:** The Norwegian Research Council (agreement 259293) funded this project. The funders had no role in study design, data collection and analysis, decision to publish, or preparation of the manuscript. SINTEF Digital provided support in the form of salaries for author [EG-C], but did not have any additional role in the study design, data collection and analysis, decision to publish, or preparation of the manuscript. Part of the authors [EG-C] work was done when he was former employed at the University of Oslo; all work done after he joined SINTEF Digital has been done on the authors' free time. SINTEF Digital is not paying for this study. The specific roles of the authors are articulated in the 'author contributions' section.

**Competing interests:** SINTEF Digital provided support in the form of salaries for author [EGC]. This does not alter our adherence to PLOS ONE policies on sharing data and materials. The other authors have no competing interests.

# Introduction

The current practice of assessing mood episodes in affective disorders are subjective observations combined with semi-structured clinical rating scales. Objective methods for assessing affective symptoms are desired [1]. Motor activity is an objective observation of the inner physiological state expressed in behavior patterns, and alterations in activation are essential symptoms of bipolar and unipolar depression [2, 3]. The depressive state is typically associated with reduced daytime motor-activity, increased variability in activity levels and less complexity in activity patterns compared to healthy controls [2]. However, in some bipolar and unipolar depressed patients contradictory motor activity patterns have been observed, characterized by increased mean activity levels, reduced variability and an augmented complexity in activity patterns more similar to that observed in manic patients [4]. Such depressions are commonly associated with irritability, restlessness, and aroused inner tension, in contrast to the general loss of initiative and interest characterizing psychomotor retarded depressions [5].

It has been suggested by Sabelli et al. [6] that mood disorders are diseases of energy fluctuations, and a thermodynamic model of bipolar disorder has been proposed. Simplified the model represents two energies emanating out of a mutual zero point of down-regulated motor retarded depression. The first euphoric energy represents arousal of manic symptoms like inflated self-esteem and increased goal-directed actions. The second agitated energy is associated with aroused inner tension, anxiety and restlessness. The euthymic condition oscillates within a healthy range on both energies. There is evidential support for the thermodynamic hypothesis as amplified levels of euphoric and agitated energy seems present within the manic state [7], and agitated energy seems present in approximately one out of five depressions, regardless of polarity [8].

Motor activity is indisputably an articulation of repeated daily social rhythms in interaction with cyclical biological rhythms, driven by the 24-hour circadian clock interlocked with numerous ultradian rhythmic cycles of 2 to 6 hours [9]. Out of sync biological rhythmic patterns are suggested as essential symptoms of mood episodes [10]. Time series of recurring biological rhythms and day-to-day life patterns are to be considered as complex dynamical systems [11]. Complex dynamical systems rarely categorizes by simple linear models. Therefore, mathematical tools obtained from the field of non-linear complex and chaotic systems have been the traditional method for analyzing and evaluating motor activity recordings [12–14]. Machine learning (ML) techniques have displayed promising results in analyzing data of complex dynamical systems [15, 16], and MLs ability to reveal non-obvious patterns has fairly accurately classified mood state in long-term heart rate variability analysis of bipolar patients [17]. Nonlinear heart rate variability analyses have similarly identified altered cardiovascular autonomic functions in manic patients [18]. Accelerometer recordings are considerably more noisy than heart rate data [19]. Still, motor activity time series hold prodigious potential for various ML approaches. Techniques like Random Forest [20] and neural networks [21, 22] have revealed promising abilities to handle time series of activation data.

A neural network might be understood as a mathematical model, where millions of parameters automatically fine-tunes to optimize the models' performance [23, 24]. Consequently, insight into the lines of argument is difficult. However, there are methods that allow the interpretation of neural network internals to some extent [25]. Within medical science, there is skepticism of such a black-box method generating calculations without an explanation [26]. However, outcomes from analyses of essential variables of high quality ought to be considered trustworthy, at least when measures to counteract overfitting have been applied [27]. The ensemble learning method of the Random Forest algorithm is robust against overfitting, and the approach might be understood as a woodland of decision trees, where multiple trees look

at stochastic parts of the data [28]. Decision trees' decisions are transparent, and lines of argument interpretable [29].

The aim was to investigate if objective biological measures can aid existing diagnostic practice, by applying machine-learning techniques to analyze motor activity patterns from depressed patients and healthy controls.

## Materials and methods

### Sample characteristics

This is a reanalysis of motor activity recordings originating from an observational cohort study presented in previous papers [12, 13, 30]. The study group consisted of 23 bipolar and unipolar outpatients and inpatients at Haukeland University Hospital, Bergen, Norway. All fulfilled the criteria for a major depression, according to a semi-structured interview based on DSM-IV criteria for mood disorders [31]. The severity of the depressive symptoms was evaluated on the Montgomery and Aasberg Depression Rating Scale (MADRS) at the beginning and conclusion of the motor-activity recordings [32]. 15 of the patients were on antidepressants, some co-medicated with either mood stabilizers or antipsychotics, the rest did not use any psychiatric medications. Further description of the study group is presented in previous papers.

The control group consisted of 32 heathy individuals, all without a history of either psychotic or affective disorders. The majority were shift working hospital staffs. Both datasets are available for other researchers [33]. The Norwegian Regional Medical Research Ethics Committee West approved the study protocol, a written informed consent was obtained from all participants involved in the study, no compensations for participating in the study were given, and all processes were in accordance with the Helsinki Declaration of 1975.

### Recording of motor activity

Motor activity was recorded with a wrist-worn actigraph entailing a piezoelectric accelerometer programmed to record the integration of intensity, amount and duration of movement in all directions. The sampling frequency was 32 Hz and movements over 0.05 g recorded. The output was gravitational acceleration units per minute [30]. The Actigraph device was worn continuously throughout the complete recording period.

### Machine learning

The basic framework of our ML approach has earlier been presented in a technological conference paper [34], but the method presented here represents a substantial extension of the previous work. Given that the main objective was to classify a user as depressed or not depressed, we proposed the following approach to accomplish this: Each user collected data for $d_i$ consecutive days where $d_i$ represents the number of days collected by participant $i$. Then, statistical features capturing overall activity levels and variations from each day were extracted [35], resulting in $d_i$ feature vectors per participant, and then normalized per participant to values between zero and one. That is, the normalization parameters (max and min values) were learned from the training set users. The features were extracted in the statistical software R version 3.6.0.

To avoid overfitting, we adopted a Leave-One-User-Out validation strategy, i.e., for each user $i$, use all the data from all other *users not equal to i* to train the classifier and test them using the data from user $i$. In order to obtain the final classification for a particular user, depressed or not depressed, a vector of predictions **p** is first obtained from the trained

classifier. Each entry of **p** corresponds to the prediction of a particular day. The final label was obtained by majority voting, i.e., output the most frequent prediction from **p** [27].

Our dataset was imbalanced with 291 depressed and 402 not depressed states, yet it is regarded as a realistic representation of real-world clinical data [36]. As ML algorithms generally have a tendency to favor the most represented class, we tested two different class balancing oversampling techniques for augmenting the minority class [37]. Firstly, we used random oversampling, which duplicates data points selected at random. Secondly, we used SMOTE [38], which creates new synthetic samples that are generated at random from similar neighboring points. Furthermore, we tested three different machine learning classifiers, Random Forest [39], unweighted and weighted Deep Neural Network (DNN) and a weighted Convolutional Neural Network (CNN) [40]. The weighted DNN and CNN use class weights at training time to weight the loss function. The weight for the depressed class was set as *wdepressed = α/β* where α is the number of instances that belong to the majority class (not depressed) and β are (or maybe denotes) the number of points that belong to the minority class (depressed). The weight for the not depressed class was set as *wnondepressed = α/α = 1*. This weighting informs the algorithm to pay more attention to the underrepresented class. The weighting parameters were learned from the training set. For the weighted approaches, neither random oversampling nor SMOTE were utilized as this will be double compensating for the class imbalance in the training set.

Random forest is an ensemble method that uses multiple learning models to gain better predictive results. It consists of several decision trees. Each decision tree considers a subset of features to solve the problem at hand. Each subset has only access to a subset of the training data points, consequently leading to a more robust overall performance by increasing the diversity in the forests. The subsets are chosen randomly, and the final prediction is an average from all sub decision trees within the forest [39]. The code was implemented in the statistical software R with the use of the *randomForest* library [28].

The DNN architecture consisted of two fully connected hidden layers with 128 and 8 units respectively with a rectified linear unit (ReLU) as activation function. After each layer, we applied dropout (p = 0.5) and the last layer has 2 units with a softmax activation function. The CNN architecture entailed two convolutional layers where max pooling and dropout (p = 0.25) were used. Then, two more convolutional layers also applying max pooling and dropout (p = 0.25) followed. Lastly, the data was flattened, and there was a fully connected layer of 512 units with dropout (p = 0.50). Each convolutional layer had a kernel of size 3 with a stride size of 1. The number of kernels for the first two convolutional layers was 16, and 32 for the last 2 layers. The max pooling size was 2. The activation functions of the convolutional layers and the fully connected layer were ReLUs. Finally, a fully connected layer with 2 units and softmax activation function was used to produce the prediction [40]. For the CNN, instead of extracting features, we represented each day as an image with 24 rows and 60 columns. The rows represent the hour of the day, and the columns represent the minute for each particular hour. Each entry is the activity level registered by the device. Missing values were filled with -1, which accounted for 3.6% of all data points. The motivation of using the CNN approach, was the preservation of more information compared to a feature-based approach. With the CNN and the chosen representation, the granularity is at the minute level and there is no to need do feature extraction. On the other hand, for the non-CNN based methods, the features were computed on a daily basis, which can lead to some information being lost. Both networks trained for 30 epochs with a batch size of 32. The code was written in R (version 3.6.0) using the *Keras* library with Tensorflow 1.13 as the backend. For baseline classifier, we used a classifier that outputs a random class only based on their prior probabilities regardless of the input data. Note that the baseline predictions were computed separately for each of the machine

learning methods. Since the baseline was based on random predictions, results may vary slightly across re-runs.

## Statistics

The intention of statistical feature extraction from the raw data file is to distillate the dataset into a few variables adaptable for the machine learning algorithms, ideally capturing the essential content of the original dataset [27]. As no established practice exists, a common way to find out what features to select for a given dataset is empirically evaluating different features [41]. The statistical features extracted for this experiment were mean activity level, the corresponding standard deviation (SD) and the proportion of minutes with an activity level of zero. The estimates were chosen due to previous experiences in analyzing accelerometer data with ML [42]. Mean values were calculated from the pre-normalized features per day for each participant, and significance tested with SPSS version 24. Independent Samples T-Test with Levene's Test of Equality of Variances were applied when comparing two groups. One-way ANOVA when comparing more than two groups, followed by Bonferroni corrections to evaluate pairwise differences between groups. A p-value less than 0.05 was considered statistically significant.

## Outcome metrics

Since our ML objective was to classify cases as either depressed mood or controls, the outcome of machine learning algorithms were given in measures of accuracy for binary tests [43]. *Sensitivity* is the fraction of correctly classified conditions related to all conditions and *specificity* the fraction of controls correctly classified as controls. *Weighted recall* is an estimate combining sensitivity and specificity equalized according to sample sizes. The *positive* (PPV) and *negative* (NPV) *predictive values* represent the amount of correct classifications related to the amount of wrong classifications of either conditions (positive) or controls (negative). *Weighted precision* is an estimate combining the predictive values according to sample sizes. Although the estimate Accuracy is a common indicator when reporting outcomes, it does not consider imbalance in the dataset, and therefore potentially presents misrepresentative outcomes. For evaluating the overall performance of the ML classifiers, we used the *Matthews Correlation Coefficient* (MCC) that is recommended when datasets are imbalanced [44]. MCC gives a coefficient value between minus one and one, and zero indicates a random estimation.

For the interpretability analysis we used the model-agnostic method Partial Dependence Plots (PDPs) to illustrate separately each of the extracted statistical features' impact on the Random Forest outcome [45]. To generate the plots, the *pdp* R library was used [46]. Classes were converted to numeric: depressed = 1 and control = 0, and the partial dependence of a set of features of interest *zs* was estimated by averaging the predictions for each unique value of *zs* while keeping the other variables fixed [29].

## Results

The condition group analyzed in the first ML runs consisted of 10 females and 13 males, aged 42.8 years (standard derivation (SD) = 11 years), and with average actigraph recordings of 12.7 days (SD = 2.8, range 5–18 days). Mean MADRS score at the start of registrations was 22.7 (SD = 4.8), and at the end 20.0 (SD = 4.7). Fifteen persons were diagnosed with unipolar depression and eight with bipolar disorder. The control group consisted of 20 females and 12 males, average age was 38.2 (SD = 13), and the group wore the actigraph for an average of 12.6 days (SD = 3.3, range 8–20) (Table 1).

**Table 1. Characteristics of the depressed patients and healthy controls analyzed in the first machine learning run.**

| | Depressed patients | Healthy Controls | t-test* |
|---:|:---:|:---:|:---:|
| Label | Condition | Control | |
| Days[1] | 291 | 402 | |
| N | 23 | 32 | |
| Gender (male/female) | 13 / 10 | 12 / 20 | |
| Age | 42.8 (11.0) | 38.2 (13.0) | p = 0.170 |
| Days used Actigraph | 12.7 (2.8) | 12.6 (2.3) | p = 0.897 |
| Diagnosis (unipolar/bipolar) | 15 / 8 | | |
| MADRS at start | 22.7 (4.8) | | |
| MADRS at end | 20.0 (4.7) | | |
| *Extracted Statistical Features*: | | | |
| Mean Activity | 190.05 (81.44) | 286.59 (81.10) | ***p < 0.001*** |
| SD[2] | 300.54 (95.86) | 405.10 (99.7) | ***p < 0.001*** |
| Proportion of Zeros[3] | 0.385 (0.154) | 0.299 (0.086) | ***p = 0.010*** |

All data are given as mean (standard derivation), if not otherwise specified.

* Independent Samples T-Test with Levene's Test of Equality of Variances, significance level $p < 0.05$.

[1] Number of 24-h sequences analyzed.

[2] Standard derivation of mean activity.

[3] Ratio of minutes with an activity level of zero.

The best performing ML algorithm in the first run was the weighted CNN approach, correctly classifying 65% of the depressed patients as conditions (sensitivity: 0.65), with a true negative rate of 0.78 (specificity). The overall capabilities of the algorithm to classify both depressions and controls were 0.73 (weighted recall), and overall performance was 0.44 (MCC). Class balancing techniques improved the performance of both of the unweighted ML approaches. Random Forest with SMOTE oversampling technique achieved a sensitivity of 0.61, weighted recall of 0.69 and a MCC of 0.36. In the DNN experiments, random oversampling performed best, with a sensitivity of 0.52, weighted recall of 0.69 and a MCC of 0.35, even though SMOTE achieved a higher sensitivity (0.57). The weighted DNN approach achieved a weighted recall of 0.65 and a MCC of 0.29 (Table 2).

The interpretability analysis of the Random Forest classifier is presented in a partial dependence plot for each analyzed feature (Fig 1). Regarding the mean activity level and standard deviation of mean activity, the overall tendency was decreasing values associated with a condition classification. The trend differs for the proportion of zero activity, where increasing percentage was associated with condition predicted as outcome. Similar overall tendencies were statistical observable between the groups (Table 1), as the depressed patients were significantly lower in mean activity ($p < 0.001$) and SD of mean activity ($p < 0.001$), and had elevated ratios of minutes with an activity level of zero ($p = 0.010$) compared to controls.

As an attempt to capture the quintessence of the misclassified groups, the false cases were commonly identified as misclassifications in the five previous mentioned ML algorithms; weighted CNN, weighted DNN, DNN (SMOTE and random oversampling) and Random Forest (SMOTE). Six conditions were constantly classified falsely (FN) in all five outcomes. There were no significant differences (t-test) when comparing FN and the correctly classified depressions (TP) on MADRS scores. Fewer controls were commonly misclassified (n = 4). For that reason, the false positives group (FP) consisted of all controls misclassified in at least three out of five predicted outcomes (n = 7). When comparing all four outcome groups, FN, FP, TP and true negative controls (TN), we found ANOVA significant group differences for all the

**Table 2. Machine learning classification results (1st run) in motor activity time series from depressed patients (n = 23) and healthy controls (n = 32).**

| Machine Learning Approach | Class Balancing Technique | Classification results by label | | | | | | | | | | | |
|---|---|---|---|---|---|---|---|---|---|---|---|---|---|
| | | Sensitivity | Specificity | Weighted Recall | PPV | NPV | Weighted Precision | Accuracy | MCC | TP | TN | FP | FN |
| | Baseline | 0.22 | 0.75 | 0.53 | 0.38 | 0.57 | 0.49 | 0.53 | −0.4 | 5 | 24 | 8 | 18 |
| **Random Forest** | No oversampling | 0.44 | 0.75 | 0.62 | 0.56 | 0.65 | 0.61 | 0.62 | 0.19 | 10 | 24 | 8 | 13 |
| | Random oversampling | 0.52 | 0.75 | 0.65 | 0.60 | 0.69 | 0.65 | 0.65 | 0.28 | 12 | 24 | 8 | 11 |
| | SMOTE | **0.61** | 0.75 | **0.69** | 0.64 | 0.73 | 0.69 | 0.69 | **0.36** | 14 | 24 | 8 | 9 |
| | Baseline | 0.22 | 0.88 | 0.60 | 0.56 | 0.61 | 0.59 | 0.60 | 0.12 | 5 | 28 | 4 | 18 |
| **Deep** | No oversampling | 0.43 | 0.84 | 0.67 | 0.67 | 0.68 | 0.67 | 0.67 | 0.31 | 10 | 27 | 5 | 13 |
| **Neural Network** | Random oversampling | **0.52** | 0.81 | **0.69** | 0.67 | 0.70 | 0.69 | 0.69 | **0.35** | 12 | 26 | 6 | 11 |
| | SMOTE | **0.57** | 0.75 | 0.67 | 0.62 | 0.71 | 0.67 | 0.67 | 0.32 | 13 | 24 | 8 | 10 |
| **Weighted Deep** | Baseline | 0.22 | 0.88 | 0.60 | 0.56 | 0.61 | 0.59 | 0.60 | 0.12 | 5 | 28 | 4 | 18 |
| **Neural Network** | No oversampling | **0.61** | 0.69 | **0.65** | 0.58 | 0.71 | 0.66 | 0.65 | **0.29** | 14 | 22 | 10 | 9 |
| **Weighted Convolutional** | Baseline | 0.35 | 0.59 | 0.49 | 0.38 | 0.56 | 0.48 | 0.49 | −0.06 | 8 | 19 | 13 | 15 |
| **Neural Network** | No oversampling | **0.65** | **0.78** | **0.73** | 0.68 | 0.76 | 0.73 | 0.73 | **0.44** | 15 | 25 | 7 | 8 |

TP: True Positives (condition cases classified correctly as labeled).

FN: False Negatives (condition cases misclassified as control cases).

TN: True Negatives (control cases classified correctly as labeled).

FP: False Positives (controls cases misclassified as condition cases).

Sensitivity: True Positive Rate; TP / (TP + FN). Specificity: True Negative Rate; TN / (TN + FP). Weighted Recall: (Sensitivity x (TP + FN)) + (Specificity x (TN + FP)) / (TP + FN + TN + FP). PPV: Positive Predictive Value; TP / (TP + FP). NPV: Negative Predictive Value: TN / (TN + FN). Weighted Precision: (PPV x (TP + FN)) + (NPV x (TN + FP)) / (TP + FN + TN + FP). Accuracy: (TP + TN) / (TP + TN + FP + FN). MCC: Matthews Correlation Coefficient: ((TP x TN)–(FP x FN)) / sqrt ((TP + FP) x (TP + FN) x (TN + FP) x (TN + FN)).

analyzed statistical features (p < 0.001) (Table 3). There were no significant group differences (ANOVA) for age composition and the number of days the participants wore the Actigraph.

As shown in Table 3 the TP conditions have Bonferroni significantly reduced mean activity compared to both FN conditions (p = 0.002) and TN controls (p < 0.001). The TP conditions

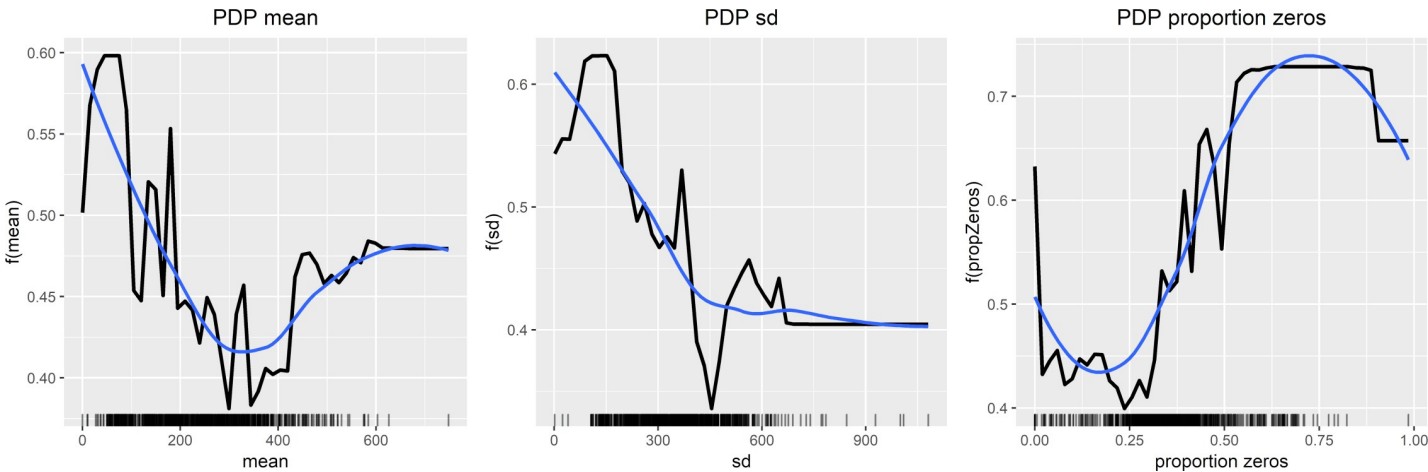

**Fig 1. Model interpretability analysis (1st run).** Partial Dependence Plots (PDPs) of the Random Forest classification. The x-axis represents the feature value whereas the y-axis is the models output value.

**Table 3. Characteristics of classification results by predicted condition from the 1st machine learning run.**

| | Predicted groups | | | | ANOVA* |
|---|---|---|---|---|---|
| | True Positives | False Negatives | True Negatives | False Positives | |
| Label | Condition | Condition | Control | Control | |
| N (Days[1]) | 17 (219) | 6 (72) | 25 (309) | 7 (93) | |
| Mean Activity | 159.68 (68.34)[a/b] | 276.09 (47.10)[b] | 313.31 (68.08)[a/c] | 191.16 (42.94)[c] | $F(3,51) = 21.76$, $p < 0.001$ |
| SD[2] | 263.61 (79.22)[a/d] | 405.18 (50.75)[d] | 433.63 (91.29)[a/e] | 303.23 (51.92)[e] | $F(3,51) = 16.93$, $p < 0.001$ |
| Proportion of Zeros[3] | 0.435 (0.106)[f/d] | 0.245 (0.191)[d] | 0.295 (0.064)[f] | 0.312 (0.147) | $F(3,51) = 7.59$, $p < 0.001$ |

All data are given as mean (standard derivation), if not otherwise specified.

* One-way ANOVA, significance level p < 0.05.

[1] Number of 24-h sequences.

[2] Standard Derivation of mean activity.

[3] Ratio of minutes with an activity level of zero.

Post hoc Bonferroni tests (significance level p < 0.05).

[a] p < 0.001—TP compared to TN.

[b] p = 0.002—FN compared to TP.

[c] p < 0.001—FP compared to TN.

[d] p = 0.003—FN compared to TP.

[e] p = 0.002—FP compared to TN.

[f] p = 0.001—TP compared to TN.

had also Bonferroni significant decreased SD compared to FN conditions (p = 0.003) and TN controls (p < 0.001). Furthermore, the portions of minutes with an activity level of zero were significantly higher for TP compared to FN (p = 0.003) and TN (p = 0.001). There were no Bonferroni significant differences between the misclassified depressions (FN) and the two control groups (TN + FP), but the misclassified control group (FP) had significantly reduced mean activity (p < 0.001) and SD (p = 0.002) compared to TN controls.

For the second ML runs, the six patients identified as the FN condition group of the first ML runs were omitted from the analysis. This time the condition group consisted of 7 females and 10 males, with an average age of 45.2 (SD = 10.4) years, and with average actigraph recordings of 12.7 (SD = 3.0, range 5–18) days. Eleven were diagnosed with unipolar depression and six with bipolar disorder (Table 4).

DNN with SMOTE oversampling technique performed unmatched with an overall weighted accuracy (MCC) of 0.65, a true positive rate of 0.82 (sensitivity), a true negative rate of 0.84 (specificity) and a weighted recall of 0.84. Random oversampling DNN achieved a sensitivity of 0.82, weighted recall of 0.80 and a MCC of 0.58, and weighted DNN performed with a similar sensitivity, a weighted recall of 0.78 and a MCC of 0.55. Overall, the DNN approaches performed a cut above the rest, as negative predictive values (NPV) around 0.90 indicates a limited number of depressions incorrectly classified as controls. The best performing Random Forest approach (SMOTE) achieved a MCC of 0.53 and a weighted recall of 0.78. Weighted CNN performed with a MCC of 0.46 and weighted recall of 0.76 (Table 5).

To illustrate the characteristics of the misclassified condition and control groups of the second ML runs, we looked to the misclassifications of the DNN approaches that performed a cut above the rest. Three condition and five control cases were commonly falsely classified. When comparing the four outcome groups (TP, FN, TN and FP) we found statistically significant ANOVA differences (p < 0.001) for all the analyzed statistical features (Table 6). There were no significant differences (t-test) for MADRS scores, mean age and days the Actigraph was worn when comparing the TP and FN conditions groups (t-test). However, there were

**Table 4. Characteristics of depressed patients and healthy controls analyzed in the 2nd machine learning run.**

|  | Depressed patients | Healthy Controls | t-test[*] |
|---|---|---|---|
| Label | Condition | Control |  |
| Days[1] | 219 | 402 |  |
| N | 17 | 32 |  |
| Gender (male/female) | 10 / 7 | 12 / 20 |  |
| Age | 45.2 (10.4) | 38.2 (13.0) | p = 0.060 |
| Days used Actigraph | 12.9 (1.8) | 12.6 (2.3) | p = 0.624 |
| Diagnosis (unipolar/bipolar) | 11 / 6 |  |  |
| MADRS at start | 22.0 (5.1) |  |  |
| MADRS at end | 19.4 (4.6) |  |  |
| *Extracted Statistical Features*: |  |  |  |
| Mean Activity | 159.68 (68.34) | 486.59 (81.10) | ***p < 0.001*** |
| SD[2] | 263.61 (79.22) | 405.10 (99.87) | ***p < 0.001*** |
| Proportion of Zeros[3] | 0.435 (0.106) | 0.299 (0.086) | ***p < 0.001*** |

All data are given as mean (standard derivation), if not otherwise specified.

[*] Independent Samples T-Test with Levene's Test of Equality of Variances, significance level $p < 0.05$.

[1] Number of 24-h sequences analyzed.

[2] Standard Derivation of mean activity.

[3] Ratio of minutes with an activity level of zero.

significant group differences (ANOVA) for age composition when comparing the four outcome groups (F (3, 45) = 3.57, p < 0.021), but no significant differences for number of days the participants wore the Actigraph. The FN group consisted of two males and one female, one diagnosed with bipolar depression and two with unipolar. The FP group included one male and four females.

As shown in Table 6, the TP conditions presents Bonferroni significantly reduced mean activity compared to FN conditions (p = 0.014) and TN controls (p < 0.001). The FP controls presents significantly reduced mean activity levels compared to TN controls (p < 0.002). The SD of mean activity is significantly reduced for TP conditions compared to TN controls (p < 0.001), and for FP controls compared to TN controls (p = 0.016). For the proportion of minutes with an activity level of zero, TP conditions presents significantly increased portions compared to TN controls (p < 0.001). There were no Bonferroni significant differences between the misclassified depressions (FN) and the two control groups (TN + FP), as well as between the misclassified controls (FP) and the two condition groups (TP + FN).

Fig 2 presents the interpretability analysis of the second runs' Random Forest classifier, illustrating the features' behavior in the algorithm and their impact on the decisions. The overall tendencies observed in the plots look more defined and stronger than trends in the PDPs of the first ML run. This is as expected, due to the identical data analyzed with a reduced condition group. Similar to the observed significant group differences presented in Table 4, decreasing values of mean activity and standard deviation of mean activity, as well as increasing proportion of zero activity associates with the condition state classification. Furthermore, these trends reflect the significant differences between the four groups presented in Table 6.

## Discussion

In our quest to answer the objective of the study, we have tested three different machine learning classifiers analyzing an imbalanced dataset with a larger control group than condition

**Table 5. Machine learning classification results (2nd run) in motor activity time series of motor retarded depressed patients (n = 17) and healthy controls (n = 32).**

| Machine Learning Approach | Class Balancing Technique | Classification results by label | | | | | | | | | | | | |
|---|---|---|---|---|---|---|---|---|---|---|---|---|---|---|
| | | Sensitivity | Specificity | Weighted Recall | PPV | NPV | Weighted Precision | Accuracy | MCC | TP | TN | FP | FN |
| | Baseline | 0.18 | 0.75 | 0.55 | 0.27 | 0.63 | 0.51 | 0.55 | −0.08 | 3 | 24 | 8 | 14 |
| **Random Forest** | No oversampling | 0.47 | 0.84 | 0.71 | 0.62 | 0.75 | 0.70 | 0.71 | 0.34 | 8 | 27 | 5 | 9 |
| | Random oversampling | 0.65 | 0.81 | 0.76 | 0.65 | 0.81 | 0.76 | 0.76 | 0.46 | 11 | 26 | 6 | 6 |
| | SMOTE | **0.76** | 0.78 | **0.78** | 0.65 | 0.86 | 0.79 | 0.78 | **0.53** | 13 | 25 | 7 | 4 |
| | Baseline | 0.06 | 0.91 | 0.61 | 0.25 | 0.64 | 0.51 | 0.61 | −0.06 | 1 | 29 | 3 | 16 |
| **Deep** | No oversampling | 0.53 | 0.91 | 0.78 | 0.75 | 0.78 | 0.77 | 0.78 | 0.48 | 9 | 29 | 3 | 8 |
| **Neural Network** | Random oversampling | **0.82** | 0.78 | **0.80** | 0.67 | **0.89** | 0.81 | 0.80 | **0.58** | 14 | 25 | 7 | 3 |
| | SMOTE | **0.82** | **0.84** | **0.84** | 0.74 | **0.90** | 0.84 | 0.84 | **0.65** | 14 | 27 | 5 | 3 |
| **Weighted Deep** | Baseline | 0.06 | 0.88 | 0.59 | 0.20 | 0.64 | 0.49 | 0.59 | −0.10 | 1 | 28 | 4 | 16 |
| **Neural Network** | No oversampling | **0.82** | 0.75 | **0.78** | 0.64 | **0.89** | 0.80 | 0.78 | **0.55** | 14 | 24 | 8 | 3 |
| **Weighted Convolutional** | Baseline | 0.18 | 0.78 | 0.57 | 0.30 | 0.64 | 0.52 | 0.57 | −0.05 | 3 | 25 | 7 | 14 |
| **Neural Network** | No oversampling | **0.65** | 0.81 | **0.76** | 0.65 | 0.81 | 0.76 | 0.76 | **0.46** | 11 | 26 | 6 | 6 |

TP: True Positives (condition cases classified correctly as labeled).

FN: False Negatives (condition cases misclassified as control cases).

TN: True Negatives (control cases classified correctly as labeled).

FP: False Positives (controls cases misclassified as condition cases).

Sensitivity: True Positive Rate; TP / (TP + FN). Specificity: True Negative Rate; TN / (TN + FP). Weighted Recall: (Sensitivity x (TP + FN)) + (Specificity x (TN + FP)) / (TP + FN + TN + FP). PPV: Positive Predictive Value; TP / (TP + FP). NPV: Negative Predictive Value: TN / (TN + FN). Weighted Precision: (PPV x (TP + FN)) + (NPV x (TN + FP)) / (TP + FN + TN + FP). Accuracy: (TP + TN) / (TP + TN + FP + FN). MCC: Matthews Correlation Coefficient: ((TP x TN)–(FP x FN)) / sqrt ((TP + FP) x (TP + FN) x (TN + FP) x (TN + FN)).

**Table 6. Characteristics of classification results by predicted condition from the second machine learning run.**

| | Predicted groups | | | | |
|---|---|---|---|---|---|
| | True Positives | False Negatives | True Negatives | False Positives | ANOVA[*] |
| Label | Condition | Condition | Control | Control | |
| N (Days[1]) | 14 (176) | 3 (43) | 27 (335) | 5 (67) | |
| Mean Activity | 136.78 (50.18)[a/b] | 266.47 (13.71)[b] | 304.90 (74.05)[a/c] | 187.70 (29.06)[c] | $F_{(3,45)} = 23.32$, **p < 0.001** |
| SD[2] | 239.59 (64.38)[a] | 375.70 (19.56) | 424.79 (95.10)[a/d] | 298.77 (41.90)[d] | $F_{(3,45)} = 16.93$, **p < 0.001** |
| Proportion of Zeros[3] | 0.454 (0.107)[a] | 0.347 (0.038) | 0.283 (0.084)[a] | 0.383 (0.026) | $F_{(3,45)} = 12.33$, **p < 0.001** |

All data are given as mean (standard derivation), if not otherwise specified.

[*] One-way ANOVA, significance level $p < 0.05$.

[1] Number of 24-h sequences.

[2] Standard Derivation of mean activity.

[3] Ratio of minutes with an activity level of zero.

Post hoc Bonferroni tests (significance level $p < 0.05$).

[a] $p < 0.001$—TP compared to TN.

[b] $p = 0.014$—FN compared to TP.

[c] $p < 0.002$—FP compared to TN.

[d] $p = 0.016$—FP compared to TN.

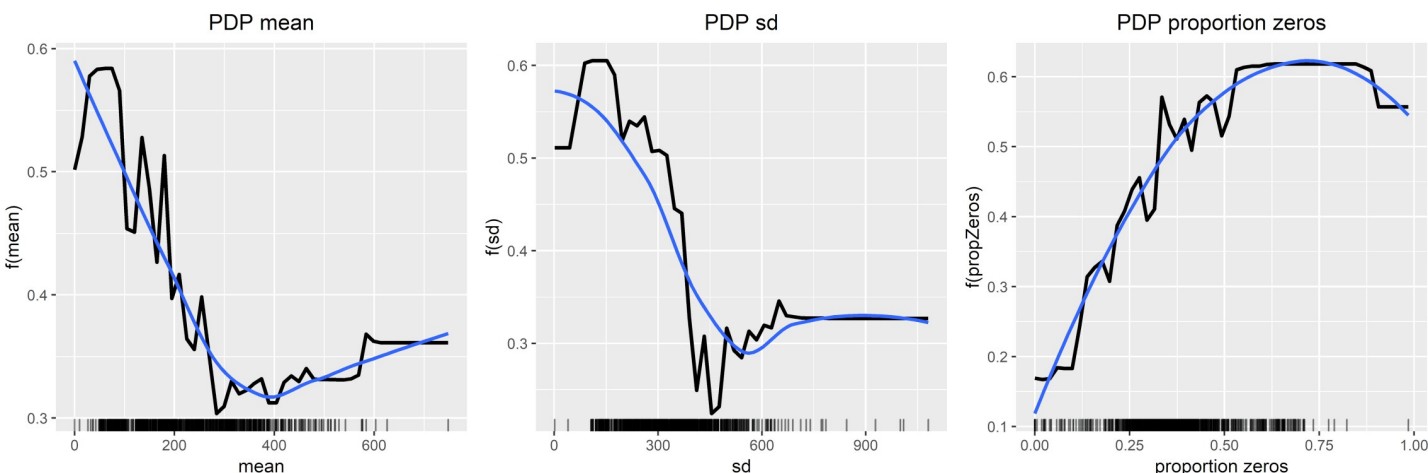

**Fig 2. Model interpretability analysis (2nd run).** Partial Dependence Plots (PDPs) of the Random Forest classification. The x-axis represents the feature value whereas the y-axis is the models output value.

group. ML´s ability to discriminate between depressed patients and healthy controls seems generally promising, as our results are substantially above both random and the baseline predictions. According to our experiment, the Deep Neural Network algorithm seems to be the preeminent ML approach to discriminate between depressed patients and healthy controls in motor activity data. The most optimistic overall result was attained by DNN with the SMOTE oversampling technique, classifying 82% of the depressed patients correctly and 84% of the controls. Random oversampling DNN and weighted DNN also coped with classifying 82% of the depressions correctly, but achieved only to classify respectively 78% and 75% of the controls correctly. However regardless of the outcome, all machine learning approaches utilized in this experiment achieved better results than what was found in a previous study employing a nonlinear discriminant function statistical analysis to differentiate between mood states in 24 hours motor activity recordings of bipolar inpatients [14]. This method managed to classify manic and mixed states rather accurately, but 42% of the depressions misclassified as manic. The misclassification is explained by the fact that depressed patients appear to be a complex and varied group, as some patients presents manic-like motor activity patterns in their depressive state. Therefore, it is more appropriate to compare this study's results with the results of our experiment's first ML run. In the second ML run, the depressed patients misclassified in the first ML run were omitted from analyzes, as their motor activity patterns looked more analogous to the correctly classified controls, and seem to have contributed only as noisy confusion in the first analysis. Regarding the manic-like patterns, it is previously demonstrated that the motor activity patterns of mania look more similar to those of healthy controls than depressive patients [2].

The rationale behind doing a second ML run omitting the misclassification was not to make the classification problem easier. As the purpose of this study was to investigate if activity measures can aid clinical diagnostics, we needed to recognize the possible presence of agitation in depression due to the significant differences in motor activity patterns. A study by Krane-Gartiser and colleagues [4] investigated group differences between unipolar depressed inpatients with and without motor retardation in 24 hours of motor activity recordings. Depressions without motor retardation were found to have increased mean activity levels; reduced variability compared to the motor retarded patients, as well as augmented complexity in activity patterns. The left out patients of this experiment had significantly increased mean activity

compared to the patients involved in the second analysis. The variability measure reported by this previous study is the coefficient of variation (CV) [4], a ratio estimated from SD divided by the mean. Both CV and SD are estimates of variance, expressing the stability of the mean in a time series [47]. Based on the numbers presented in Table 3, the excluded patient group (FN) appears to have reduced CV compared to the others patients (TP). Furthermore, previous studies of the complete current study population have identified increased daytime variability in activity levels for the depressed patients compared to the controls, as well as reduced complexity in daytime activity for the depressed patients [13]. Sample entropy, the nonlinear index of complexity, was not estimated for this experiment. According to experiences within this research group, it is problematic to estimate sample entropy from time series containing longer durations of zero activity, like 24-hour sequences. All series become distinctly ordered, and the ability to differentiate between groups turn out to be significantly reduced. Consequently, sample entropy is applicable only for ultradian periods of more or less continuous activity. Nevertheless, overall it is a reasonable assumption that the left out patients of this experiment are quite similar to the group of unipolar patients without motor retardation reported on by Krane-Gartiser et al.

We have investigated a heterogeneous patient group consisting of unipolar and bipolar depressed inpatients and outpatients. No differences between depressed inpatients and outpatients have previously been identified [48]. According to previous studies, unipolar and bipolar depressions seem to resemble each other in 24-hour motor activity [4, 49]. On the other hand, psychomotor restlessness appears associated with bipolar depression, and the feature seems to differentiate bipolar disorder from unipolar depressions [50]. In our sample, this does not seem to be the case as the ratio of unipolar and bipolar cases equally distributes in both the analyzed and left out patients groups. Also, the proportion of omitted depressions without motor retardation harmonizes with existing evidence as agitated energy seems present in approximately one out of five depressions [8]. Consequently, our promising ML results then only derivate from comparing the activity patterns of motor retarded depressed patients to controls.

Previous studies on accelerometer data and mood disorders have advised against applying ML in smaller samples, mainly due to the risk of overfitting producing untrustworthy results [14]. Overfitting is a phenomenon occurring when an ML algorithm trains itself to perfect predictions on a specific dataset, but then predicts poorly on new data due to systematic bias incorporated in the judgement model. In our experiments, we applied the Leave-One-User-Out validation strategy to minimize the risk of overfitting as recommended. In addition, our findings are in line with existing knowledge. Therefore, our results may be regarded as credible.

The most optimistic individual specificity result accomplished by DNN in the second ML run represents most likely the ML algorithms' tendency to favor the majority class. So when DNN without oversampling achieved a specificity (true negative rate) of 0.91 by analyzing the unprocessed and imbalanced datasets, the sensitivity was only 0.53. The most optimistic discriminating DNN approaches applied either oversampling techniques or weighting to deal with the imbalance problem. Still, weighted precision outcomes between 0.80 and 0.84 indicates a substantial number of misclassifications, probably related to data overlapping among the falsely classified classes demonstrated in Table 6. Combined, these misclassified subjects establish a gray area of intersecting activation patters, probably partly related to individual differences within the groups, as some people are natural slackers and others are born highly active. Anyhow, little is known about the control group beyond age, gender and the absence of a history of either affective or psychotic disorders. For gender, no differences in activation have previously been identified, but physical properties such as older age and higher body

mass index have previously been found to affect mean motor activity [14]. There was no information on body mass index in the dataset, but we did find age differences between the outcome groups in the second ML run. Furthermore, there was no information on external influences like seasonal time of year when the data were collected, but this has previously been considered not relevant for the evaluation of motor activity recordings [48]. On the other hand, at higher latitudes with significant seasonal change in solar insolation, hours of daylight and social rhythms, one should expect a seasonal expression in motor activity [51]. The current dataset was collected at latitude 60.4 N, a position associated with substantial seasonal change in natural light and length of day. Various psychiatric and somatic drugs may also influence motor activity patterns [14], but besides lack of medical data on the controls, the sample size is too small for such sub analyzes. Finally, being in employment could be a possible confounder of why depressed patients present lower overall activity levels than the control group.

Beside the small sample size, the main limitation of the study was the comparison between depressed patients and healthy controls. Firstly, this may have introduced errors and noise into results as individualistic divergent activity levels might have affected group differences. Secondly, although intra-individualistic comparisons suggests that the bipolar manic state is associated with increased mean activity levels, no compelling evidence has been found when comparing manic patients to healthy controls [2]. Thirdly, euthymic individuals with bipolar disorders have been found to have generally reduced mean motor activity, prolonged sleep and reduced sleep quality compared to healthy controls [52]. Therefore, a more appropriate approach would have been to compare a group of depressed patients to themselves in the euthymic state, as to identify actual alterations in motor activity related to changes in mood state [53]. To our knowledge, no such dataset exists. ML analyzes in such an intra-individualistic sample, may provide analyzes beyond the 24-h circadian cycles studied in this experience. For instance, differences between morning and evening, active and non-active periods and sleep patterns, as well as activity differences between weekdays and weekends, could be more feasible in intra-individualistic samples. In the present work, we have analyzed a possible set of statistical features extracted from activity time series, and displayed how accurate machine learning models can be trained with those. As a future direction, the utilized public dataset provides the possibility to explore the use of additional statistical features.

In conclusion, this study has illustrated the promising abilities of various machine learning algorithms to discriminate between depressed patients and healthy controls in motor activity time series. Furthermore, that the machine learning´s ways of finding hidden patterns in the data correlates with existing knowledge from previous studies employing linear and nonlinear statistical methods in motor activity. In our experiment, Deep Neural Network performed unsurpassed in discriminating between conditions and controls. Nonetheless, considering that the analyzed sample was both small and heterogeneous, we should be careful when concluding on which algorithm was the most accurate. Finally, we have enlightened the heterogeneity within depressed patients, as it is recognizable in motor activity measurements.

## Acknowledgments

This publication is part of the INTROducing Mental health through Adaptive Technology (INTROMAT) project.

## Author Contributions

**Conceptualization:** Petter Jakobsen, Ole Bernt Fasmer, Ketil Joachim Oedegaard.

**Data curation:** Petter Jakobsen, Enrique Garcia-Ceja, Michael Riegler, Ole Bernt Fasmer.

**Formal analysis:** Petter Jakobsen, Enrique Garcia-Ceja, Michael Riegler, Ole Bernt Fasmer.

**Funding acquisition:** Tine Nordgreen, Jim Torresen, Ole Bernt Fasmer.

**Investigation:** Ole Bernt Fasmer, Ketil Joachim Oedegaard.

**Methodology:** Petter Jakobsen, Ole Bernt Fasmer, Ketil Joachim Oedegaard.

**Resources:** Tine Nordgreen, Jim Torresen.

**Software:** Enrique Garcia-Ceja, Michael Riegler.

**Supervision:** Tine Nordgreen, Jim Torresen, Ole Bernt Fasmer, Ketil Joachim Oedegaard.

**Validation:** Petter Jakobsen, Enrique Garcia-Ceja, Michael Riegler, Lena Antonsen Stabell, Ole Bernt Fasmer, Ketil Joachim Oedegaard.

**Visualization:** Petter Jakobsen, Lena Antonsen Stabell, Ole Bernt Fasmer, Ketil Joachim Oedegaard.

**Writing – original draft:** Petter Jakobsen.

**Writing – review & editing:** Enrique Garcia-Ceja, Michael Riegler, Lena Antonsen Stabell, Tine Nordgreen, Jim Torresen, Ole Bernt Fasmer, Ketil Joachim Oedegaard.

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
