## [Decision Letter · Decision Letter 0]

15 May 2020

PONE-D-20-09443

Applying machine learning in motor activity time series of depressed bipolar and unipolar patients.

PLOS ONE

Dear Mr Jakobsen,

Thank you for submitting your manuscript to PLOS ONE. After careful consideration, we feel that it has merit but does not fully meet PLOS ONE’s publication criteria as it currently stands. Therefore, we invite you to submit a revised version of the manuscript that addresses the points raised during the review process. 

As you see the reviewers' comments, please particularly pay attention to the 'Materials and Methods' and 'Results' section.

We would appreciate receiving your revised manuscript by Jun 29 2020 11:59PM. To enhance the reproducibility of your results, we recommend that if applicable you deposit your laboratory protocols in protocols.io, where a protocol can be assigned its own identifier (DOI) such that it can be cited independently in the future. For instructions see: http://journals.plos.org/plosone/s/submission-guidelines#loc-laboratory-protocols

We look forward to receiving your revised manuscript.

Kind regards,

Kyoung-Sae Na, M.D.

Academic Editor

PLOS ONE

Journal Requirements:

'The authors have declared that no competing interests exist.'

We note that one or more of the authors are employed by a commercial company: SINTEF Digital.

Additional Editor Comments (if provided):

Reviewers' comments:

Reviewer's Responses to Questions

**Comments to the Author**

1. Is the manuscript technically sound, and do the data support the conclusions?

Reviewer #1: Yes

Reviewer #2: Yes

Reviewer #3: Partly

2. Has the statistical analysis been performed appropriately and rigorously? 

Reviewer #1: No

Reviewer #2: Yes

Reviewer #3: Yes

3. Have the authors made all data underlying the findings in their manuscript fully available?

Reviewer #1: Yes

Reviewer #2: Yes

Reviewer #3: Yes

4. Is the manuscript presented in an intelligible fashion and written in standard English?

Reviewer #1: Yes

Reviewer #2: Yes

Reviewer #3: Yes

5. Review Comments to the Author

Reviewer #1: The authors present the results of a classification analysis using machine learning methods to discriminate between patients with depression and healthy controls based on objective motor activity data collected with actigraph. The results show that objective activity data can be used to discriminate between patients and healthy controls with high accuracy. Using objective sensor data to diagnose and/or monitor symptoms in mental illness is both important and interesting. Several issues require consideration and should be addressed.

MAJOR ISSUES

1. In the machine learning section on line 133 the authors state that the feature vectors were normalised. It is not clear if the features were normalised per participant or across all participants. In cross-validation, the held out data should not be considered when normalising the training data to avoid learning any information from the held out data.

2. In the machine learning section on line 150-153 the authors state how the class weights are computed. However, it is not clear if the class weights are computed on the training set or across the entire dataset. In cross-validation the weights should be computed on the training set only to avoid learning from the held out data.

3. The results section on line 241-242 states “Weighted DNN performed best without class balancing techniques (no oversampling) […]” Using oversampling and class weighting at the same time will double compensate for the class imbalance in the training set and result in a biased classifier.

4. Tables 2 and 5 are not fully visible in the manuscript.

5. Figures 1 and 2 are too blurred to see axis labels and units.

6. The motivation for presenting a second run of the classification analysis without the false negatives identified in the first run is not clear. Is it not just making the classification problem easier by removing some of the “difficult” cases? The difference in mean activity between TP and FP is already demonstrated after the first run.

7. On line 466-468 the authors state that a weighted and random oversampling DNN achieves higher sensitivity and lower specificity. As stated in a previous comment, if the positive class is both weighted higher and oversampled the model is double compensating for the minority class.

8. In table 2 and 5, it is not clear why the baseline results are reported multiple times and why the results of the baselines are different every time.

MINOR ISSUES

9. In the introduction on line 94 the authors state that insight into neural networks is virtually impossible. While it may not be straight forward, there is a large research effort to improve interpretability of neural networks.

10. In the introduction on line 98-99 the authors state that “the random forest classifier is more flexible and less data-sensitive than neural networks.” It is not clear what is meant by “more flexible”. A neural networks with a non-linear activation function and a large hidden layer is a universal function approximator and thus a very flexible model.

11. In the introduction on line 101 the authors state that the random forest algorithm “has been found to predict with approximate similar quality to neural networks.” I think this is highly domain specific. While random forest is a powerful algorithm for many purposes neural networks have been proved to be superior to mostly any other method in areas such as computer vision and speech recognition.

12. It is stated on line 141 that there are 291 depressed and 402 not depressed states, but on line 150-153 ‘depressed’ is said to be majority class and ‘not depressed’ the minority class, which is contradicting.

13. Line 174-177 describes how the features are represented as an image for the CNN. It is not clear how the authors chose this feature representation or why it is appropriate for the classification task.

14. The authors already mentions limitations of comparing the patient and healthy control group. Employment status could be another significant reason why patients with depression present with lower overall activity.

Reviewer #2: This manuscript describes reanalysis of existing data applying machine-learning techniques with various data balance technique for activity patterns in depressed patients and healthy controls, and present promising abilities in discriminating between depressed patients and healthy controls in motor activity time series.

1. In the Part of Material and Methods, the detailed information of record for the motor activity might be necessary in order to achirve the integrity of the manuscript, e.g., how many time points were there in one day? How many days were recorded for each subject? Did the subjects wear the equipment all day long even during night?

2. The description of the ML process is very clear, but I am not sure if the algorithm performance is affected by depressive episode or not.

3. For the CNN, each day was represented as an image with 24 rows and 60 columns. The rows represent the hour of the day, and the columns represent the minute for each particular hour. I have two considerations:

1) I am not sure whether such new arrangement of data bring unnecessary artifact, because the data point at each minute for each hour should not have dependent relationships with a high probability. How do you explain this new data represents?

2) Missing values were filled with -1. I am interested in what distribution of these missing values for all participants? What influence may bring to the CNN?

4. From ML classification results tables, the weighted Models did not show any benefits, while, moreover, the most optimistic overall result were attained by unweighted DNN with the random oversampling technique. So, I wonder how you consider to weight for two conditions as you emphasized in particular: ‘This weighting informs the algorithm to pay more attention to the underrepresented class.’ ?

5. There must be a significant gender difference between the two groups, this might also make some data imbalance as well as recording days, how do you manage it?

6. Last, the manuscript gave one of the conclusions that Deep Neural Network performed preeminent in discriminating between conditions and controls. I suggest that we should be more careful and conservative since the sample is small and patient groups are composed by both bipolar and unipolar.

Reviewer #3: First, I reveal that I did not fully understand all the methods I applied in this study. Therefore, please read this in consideration of this.

Using this activity data through actigraphy, authors studied how to predict the depressive mood state by using various machine learning analysis methods for depression and normal control's activity. Strictly speaking, this study explores which machine learning analysis method has the best performance.

I would like to think high on how this paper tried to overcome overfitting and sample imbalance by applying various analysis techniques precisely. However, while this may simply be meaningful in terms of technical methodology, it remains fundamentally questionable as to the value of this study's hypothesis, the nature of the sample, and how valuable the research was in drawing conclusions.

It may be a good idea to use the activity level to predict the mood state of a patient with a mood disorder, but it is a very poor study in that various variables were not considered. In particular, machine learning by using the most basic values such as the mean of the activity and standard deviation causes too much to be missed. Eventually, this approach will not predict "mood depressed state", but rather predict "activity depressed state" that is supposed to be due to depression. Authors must seriously consider how to interpret and overcome this.

#1. The title appears as if the subject of this study was to differentiate motor activity in patients with bipolar and unipolar depression. It would be better to revise the subject more clearly to reveal the subject of the article.

#2. You need to use universal word in English for Keywords. It would be good to change the word into "actigraphy"

#3. Introduction Line 78~80

I agree that the time series should reflect biological rhythms and changes in daily life patterns. I think this doesn't just mean that it doesn't follow a simple linear model. It may be key to access the given activity data to fit the characteristics of the time series. What strategies did you use in this study to reflect the characteristics of your data, such as biological rhythms?

#4. Materials and methods Line 113~

The description of the sample is insufficient. Was the drug being administered at the time of the study, how long the morbidity of mood disorder was, whether receiving other nonpharmacologic treatments (eg IPSRT) that could affect the condition, were there no compensations for participating in this study, and was the study a simple observational study? If so, the criteria for inclusion and exclusion of this study should be provided. Basically, if one is depressed, he or she will not be able to comply with the study, but it is impressive that there is no significant difference in wearing days. It is necessary to calculate the wearing rate separately. In other words, it is necessary to define what wear days mean.

#5. It would not be easy to perform validation with such a small number of samples. Validation process seems to require more specific and easy to understand technology and methodology. In particular, in this case, if you repeat the learning and validation several times, the samples will eventually overlap and it may not affect the internal connectivity, which may have an impact on consequences. You need some explanation to overcome this.

#6. Please revise the figures and tables to make them readable. It is difficult to recognize.

#7. In the previous studies, authors mentioned that there was no significant difference between unipolar and bipolar depression, but I still have questions about it. Of course, since unipolar depression can be diagnosed as bipolar disorder in the future, it is not easy to make a judgment based on the current diagnosis. However, it is prudent to gather and analyze heterogeneous groups into one group. Analyzing depend on the level of activity, sampling may have its own bias. It is suggested to analyze by dividing unipolar and bipolar. And, if there is data on the normal (euthymic) mood of patients with mood disorders, not normal control people, it is necessary to compare and analyze it. It is also important to distinguish the depressed state of mood disorder from normal people, but it is more important to distinguish the euthymic and depressed mood states of mood disorders.

#8. Statistics Line 188~

How did you deal with the section that could be thought of as sleep? Did you ever think about an activity level of zero throughout the day regardless of sleep or not?

#9. Outcome Metrics 197~

You mean a depressed condition? It would be better to describe it a little more clearly. Calling a condition group is easily confusing. It is recommended to describe it a depressed mood.

#10. Table 1

Even in a healthy control group, it is basically necessary to present and compare the same psychometric values. In this study, MADRS was presented, but the results of this study do not know the mood state of the normal control group.

#11. When data related to sleep are analyzed together, some limitations of activity data can be improved. Consideration should be given to analyzing and presenting sleep data.

#12. I recommend that you try to train activity data by making it more diverse secondary variables. The strength of this data is that it is a time series. In the introduction, the characteristics of time series data and the necessity of proper analysis were explained, but in the present, only a several MLs were applied. Whether it is an analysis according to the circadian rhythm, the difference between weekdays and weekends, the difference between morning and afternoon, the difference between the most active and non-active periods, the irregularity of activities, etc. I think it is necessary. The author should considers the characteristics of the time series as much as possible, and analyzes according to the circadian rhythm, the difference between weekdays and weekends, the difference between the morning and the afternoon, the difference between the most active and non-active periods, irregularities in activities, etc. You need to do a sophisticated analysis with the possibility of creation.

6. PLOS authors have the option to publish the peer review history of their article (what does this mean?). If published, this will include your full peer review and any attached files.

Reviewer #1: No

Reviewer #2: No

Reviewer #3: No

---

## [Author Response · Author response to Decision Letter 0]

27 Jun 2020

We thank the editor and reviewers for their helpful comments that we think have contributed significantly to improving the paper. Thus, we have provided an amended manuscript with the major revisions that were recommended. Our responses can be found below, directly below the comments that they address.

Academic editor

A: Thank you for spotting these errors. We have now updated the submitted revised manuscript in accordance with PLOS ONEs requirements. All use of Symbol Font has been removed, and the title, authors and affiliations page is in accordance with the formatting guidelines. 

2. We note that one or more of the authors are employed by a commercial company: SINTEF Digital.

A: We are sorry for this misunderstanding, and have as requested updated the Funding Statement to “SINTEF Digital provided support in the form of salaries for author [EG-C], but did not have any additional role in the study design, data collection and analysis, decision to publish, or preparation of the manuscript. Part of the authors [EG-C] work was done when he was former employed at the University of Oslo; all work done after he joined SINTEF Digital has been done on the authors’ free time. SINTEF Digital is not paying for this study. The specific roles of the authors are articulated in the ‘author contributions’ section.” We have also as requested included the updated Funding Statement text in the cover letter. 

A: We have as requested updated the Competing Interests Statement. Now it reads: 

“I have read the journal’s policy, and one author of this manuscript has the following competing interests: SINTEF Digital provided support in the form of salaries for author [EG-C]. This does not alter our adherence to PLOS ONE policies on sharing data and materials. The other authors have no competing interests.”

Furthermore, we have as requested included the updated Competing Interests Statement in the cover letter. 

Reviewer #1:

The authors present the results of a classification analysis using machine learning methods to discriminate between patients with depression and healthy controls based on objective motor activity data collected with actigraph. The results show that objective activity data can be used to discriminate between patients and healthy controls with high accuracy. Using objective sensor data to diagnose and/or monitor symptoms in mental illness is both important and interesting. Several issues require consideration and should be addressed.

MAJOR ISSUES

1. In the machine learning section on line 133 the authors state that the feature vectors were normalised. It is not clear if the features were normalised per participant or across all participants. In cross-validation, the held out data should not be considered when normalising the training data to avoid learning any information from the held out data.

A: Thank you for the opportunity to clarify this, for indeed, the normalization was made across users. As suggested by the reviewer, now the normalization was done per participant to avoid learning from the test set, the experiments were rerun, and the results updated accordingly. Overall, the performance decreased a bit for all tested methods; however, the findings remain significant. We have included an improved description of how normalization was done in the machine learning section on line 137-139 “… and then normalized per participant to values between zero and one. That is, the normalization parameters (max and min values) were learned from the training set users.”

2. In the machine learning section on line 150-153 the authors state how the class weights are computed. However, it is not clear if the class weights are computed on the training set or across the entire dataset. In cross-validation the weights should be computed on the training set only to avoid learning from the held out data.

A: The class weights were learned from the training set. This has been clarified by adding, “The weighting parameters were learned from the training data” on line 161. 

3. The results section on line 241-242 states “Weighted DNN performed best without class balancing techniques (no oversampling) […]” Using oversampling and class weighting at the same time will double compensate for the class imbalance in the training set and result in a biased classifier.

A: We agree with the reviewer’s comment. This was over compensating for the minority class. Because of this, now we only include weighting with no oversampling in our experiments that include the weighted DNN and CNN. This has been described in the manuscript (line 161 - 163): “For the weighted approaches, neither random oversampling nor SMOTE were utilized as this will be double compensating for the class imbalance in the training set.”

4. Tables 2 and 5 are not fully visible in the manuscript.

A: All tables should be in line with PLOS ONEs requirements for tables, and according to instructions: “In Word, tables that run off of the manuscript page can be seen using Draft View.”

5. Figures 1 and 2 are too blurred to see axis labels and units.

A: We agree and Figures 1 and 2 are updated. 

6. The motivation for presenting a second run of the classification analysis without the false negatives identified in the first run is not clear. Is it not just making the classification problem easier by removing some of the “difficult” cases? The difference in mean activity between TP and FP is already demonstrated after the first run.

A: Thank you for the opportunity to clarify this. In the motor activity literature, to classify the mood state of depression in motor activity is considered a complicated task, as these patients appear to be a heterogeneous group, some presenting more manic-like patterns (deviating mean activity and SD). On the other hand, various agitated features seem present in about 20-25 % of all depressions. In our dataset, there was no information about agitation. For that reason, we excluded the commonly misclassified cases and labeled them as «agitated depressions». Mainly because these cases looked more like the healthy controls, e.g. presented manic-like patterns of mean activity and variability (SD). The intention for our approach was simply to enlighten the heterogeneity of activation patterns within depressed patients (as agitated energy was exhaustively presented in the introduction part). Our intention was not to make the classification problem easier by removing “difficult” cases, but we recognize that this concern should be included in our discussion in the manuscript, and have added a sentence about this (line 438 - 441): “The rationale behind doing a second ML run omitting the misclassification was not to make the classification problem easier. As the purpose of this study was to investigate if activity measures can aid clinical diagnostics, we needed to recognize the possible presence of agitation in depression due to the significant differences in motor activity patterns.”

7. On line 466-468 the authors state that a weighted and random oversampling DNN achieves higher sensitivity and lower specificity. As stated in a previous comment, if the positive class is both weighted higher and oversampled the model is double compensating for the minority class.

A: We agree. For our re-run experiments with the weighted DNN and CNN, we did not perform oversampling. The statement is therefore deleted. 

8. In table 2 and 5, it is not clear why the baseline results are reported multiple times and why the results of the baselines are different every time.

A: This is because the baseline predictions were obtained along with each of the methods. Since the baseline is based on random predictions, it will vary slightly every time it is run. The following text was added to the manuscript on line 193 - 195:

“Note that the baseline predictions were computed separately for each of the machine learning methods. Since the baseline was based on random predictions, results may vary slightly across re-runs.”

MINOR ISSUES

9. In the introduction on line 94 the authors state that insight into neural networks is virtually impossible. While it may not be straight forward, there is a large research effort to improve interpretability of neural networks.

A: We agree that that sentence was not true. We have reformulated it from "Consequently, insight into the lines of argument are virtually impossible" to "Consequently, insight into the lines of argument is difficult. However, there are methods that allow the interpretation of neural network internals to some extent (25)". (Line 94 -95)

10. In the introduction on line 98-99 the authors state that “the random forest classifier is more flexible and less data-sensitive than neural networks.” It is not clear what is meant by “more flexible”. A neural networks with a non-linear activation function and a large hidden layer is a universal function approximator and thus a very flexible model.

A: We agree that that sentence was imprecise. The sentence is now changed to “The ensemble learning method of the Random Forest algorithm is robust against overﬁtting.” The "and less data-sensitive than neural networks." was dropped, as on second thought, we do not think there is evidence to support such a strong assertion.

11. In the introduction on line 101 the authors state that the random forest algorithm “has been found to predict with approximate similar quality to neural networks.” I think this is highly domain specific. While random forest is a powerful algorithm for many purposes neural networks have been proved to be superior to mostly any other method in areas such as computer vision and speech recognition.

A: We agree that this statement is problematic, and consequently have deleted it. 

12. It is stated on line 141 that there are 291 depressed and 402 not depressed states, but on line 150-153 ‘depressed’ is said to be majority class and ‘not depressed’ the minority class, which is contradicting.

A: Thank you for spotting this error. The text has been fixed to “..the number of instances that belong to the majority class (not depressed) and β are (or maybe denotes) the number of points that belong to the minority class (depressed).” (Line 157-159)

13. Line 174-177 describes how the features are represented as an image for the CNN. It is not clear how the authors chose this feature representation or why it is appropriate for the classification task.

A: A motivation of why a CNN approach was also chosen has been included: “The motivation of using the CNN approach, was the preservation of more information compared to a feature-based approach. With the CNN and the chosen representation, the granularity is at the minute level and there is no need to do feature extraction. On the other hand, for the non-CNN based methods, the features were computed on a daily basis, which can lead to some information being lost. ” at line 185 – 189. 

14. The authors already mentions limitations of comparing the patient and healthy control group. Employment status could be another significant reason why patients with depression present with lower overall activity.

A: We agree that employment status could be a significant reason for higher overall activity, and have added the sentence: “Finally, being in employment could be a possible confounder of why depressed patients present lower overall activity levels than the control group” at line 506 – 508. This is related to a newly added sentence describing the healthy controls: “The majority of the group were shift working hospital staffs” at line 119.

Reviewer #2

This manuscript describes reanalysis of existing data applying machine-learning techniques with various data balance technique for activity patterns in depressed patients and healthy controls, and present promising abilities in discriminating between depressed patients and healthy controls in motor activity time series.

1. In the Part of Material and Methods, the detailed information of record for the motor activity might be necessary in order to achirve the integrity of the manuscript, e.g., how many time points were there in one day? How many days were recorded for each subject? Did the subjects wear the equipment all day long even during night?

A: We agree with the reviewer that this needs clarification and have added the sentence “The Actigraph device was worn continuously throughout the complete recording period.” (Line 128 – 129) in the Recording of motor activity part. We have furthermore improved the reporting on the actigraphy recordings. When reporting average day’s actigraph recordings, we have supplemented the mean and SD estimate with range (min – max). We have also updated the footnotes of tab. 1 and 4 to read, “Number of 24-h sequences analyzed” for the Days variable, as well as reported the Days variable in table 3 and 6. 

2. The description of the ML process is very clear, but I am not sure if the algorithm performance is affected by depressive episode or not.

A: This statement is based on evidence supporting that reduced daytime motor-activity and increased variability are associated with depression. Further, that alterations in motor activity might be a more defining symptom than mood for identifying depression in humans with bipolar disorder (Scott J., et al. Activation in bipolar disorders: A systematic review. JAMA Psychiatry. 2017).

3. For the CNN, each day was represented as an image with 24 rows and 60 columns. The rows represent the hour of the day, and the columns represent the minute for each particular hour. I have two considerations:

1) I am not sure whether such new arrangement of data bring unnecessary artifact, because the data point at each minute for each hour should not have dependent relationships with a high probability. How do you explain this new data represents?

A: The motivation of such an arrangement was to preserve all the information as compared to feature-extraction in which a lot of information is lost. With this image representation all the data points per day are preserved and thus, the temporal relations are captured, thus, allowing the CNN to use that information. A motivation of why this representation was used has been included in the text (line 185 – 189): “The motivation of using a CNN is because with this approach, more information is preserved as compared to a feature-based approach. With the CNN and the chosen representation, the granularity is at the minute level and there is no need to do feature extraction. On the other hand, for the non-CNN based methods, the features were computed on a daily basis which can lead to some information being lost”.

2) Missing values were filled with -1. I am interested in what distribution of these missing values for all participants? What influence may bring to the CNN?

A: The percentage of missing values has been added to the manuscript: “Missing values were filled with -1 which accounted for 3.6% of all data points” at line 185. This may have an impact on the model; however, the percentage of missing values is small.

4. From ML classification results tables, the weighted Models did not show any benefits, while, moreover, the most optimistic overall result were attained by unweighted DNN with the random oversampling technique. So, I wonder how you consider to weight for two conditions as you emphasized in particular: ‘This weighting informs the algorithm to pay more attention to the underrepresented class.’? 

A: Thank you for the opportunity to clarify this. Based on the comments from one of the reviewers (reviewer #1), we rerun the experiments without including oversampling techniques to the weighted models since this would have the effect of double compensating the minority class. 

5. There must be a significant gender difference between the two groups, this might also make some data imbalance as well as recording days, how do you manage it?

A: We agree with the reviewer that there must be a significant gender difference between the two groups of patients and controls. However, we have not looked at this topic for two reasons. Firstly, because the sample is too small to do gender sub analyzes. Secondly, because previous studies of motor activity in mood disorders have found no differences in activation between gender. The second reason is stated in the manuscript “For gender, no differences in activation have previously been identified” in line 494 and 495. 

6. Last, the manuscript gave one of the conclusions that Deep Neural Network performed preeminent in discriminating between conditions and controls. I suggest that we should be more careful and conservative since the sample is small and patient groups are composed by both bipolar and unipolar.

 A: We agree with this suggestion and have changed the conclusion to; “In our experiment, Deep Neural Network performed unsurpassed in discriminating between conditions and controls. Nonetheless, considering that the analyzed sample was both small and heterogeneous, we should be careful when concluding on which algorithm was the most accurate.” (Line 531- 534)

Reviewer #3

First, I reveal that I did not fully understand all the methods I applied in this study. Therefore, please read this in consideration of this.

Using this activity data through actigraphy, authors studied how to predict the depressive mood state by using various machine learning analysis methods for depression and normal control's activity. Strictly speaking, this study explores which machine learning analysis method has the best performance.

I would like to think high on how this paper tried to overcome overfitting and sample imbalance by applying various analysis techniques precisely. However, while this may simply be meaningful in terms of technical methodology, it remains fundamentally questionable as to the value of this study's hypothesis, the nature of the sample, and how valuable the research was in drawing conclusions.

It may be a good idea to use the activity level to predict the mood state of a patient with a mood disorder, but it is a very poor study in that various variables were not considered. In particular, machine learning by using the most basic values such as the mean of the activity and standard deviation causes too much to be missed. Eventually, this approach will not predict "mood depressed state", but rather predict "activity depressed state" that is supposed to be due to depression. Authors must seriously consider how to interpret and overcome this.

#1. The title appears as if the subject of this study was to differentiate motor activity in patients with bipolar and unipolar depression. It would be better to revise the subject more clearly to reveal the subject of the article.

A: We agree, and have changed the title of the manuscript correspondingly, “Applying machine learning in motor activity time series of depressed bipolar and unipolar patients compared to healthy controls.”

#2. You need to use universal word in English for Keywords. It would be good to change the word into "actigraphy" 

A: Good point, this has been changed. 

#3. Introduction Line 78~80

I agree that the time series should reflect biological rhythms and changes in daily life patterns. I think this doesn't just mean that it doesn't follow a simple linear model. It may be key to access the given activity data to fit the characteristics of the time series. What strategies did you use in this study to reflect the characteristics of your data, such as biological rhythms?

A: This is a limitation of our study sample. We have not applied any specific strategy to reflect the characteristics of biological rhythms in our data, beyond analyzing 24-h circadian cycles. In our opinion, to look deeper into the disrupted biological rhythmic patterns associated with depression, we would have needed intra-individual data to do an appropriately comparison of mood state differences. Anyway, we have supplemented the manuscript with the following text (line 519 - 523). “ML analyzes in such an intra-individualistic sample, may provide analyzes beyond the 24-h circadian cycles studied in this experience. For instance differences between morning and evening, active and non-active periods and sleep patterns, as well as activity differences between weekdays and weekends, could be more feasible in intra-individualistic samples”, at the end of the paragraph discussing the need for intra-individualistic data on depression. 

#4. Materials and methods Line 113~

The description of the sample is insufficient. Was the drug being administered at the time of the study, how long the morbidity of mood disorder was, whether receiving other nonpharmacologic treatments (eg IPSRT) that could affect the condition, were there no compensations for participating in this study, and was the study a simple observational study? If so, the criteria for inclusion and exclusion of this study should be provided. Basically, if one is depressed, he or she will not be able to comply with the study, but it is impressive that there is no significant difference in wearing days. It is necessary to calculate the wearing rate separately. In other words, it is necessary to define what wear days mean. 

A: We agree that the sample might appear somewhat superficially described, but as this is a reanalysis of motor activity recordings originating from a study presented in several previous papers, we have referred to these for detailed description. The original study was an observational cohort study (now mentioned in line 108) of depressed patients in traditional treatment for an ongoing major depression at a Norwegian psychiatric hospital, almost 20 years ago. There was no compensation for participating in the study and no information on duration of mood disorder. We have now inserted a new sentence into the manuscript: “no compensations for participating in the study were given” (line 122).

Since this sample is too small for sub analyzes of drug groups, we did not report on psychiatric medications. We have now inserted the sentence on line 114 - 116: “15 of the patients were on antidepressants, some co-medicated with either mood stabilizers or antipsychotics, the rest did not use any psychiatric medications.”

To define what actigraph worn day’s means, we have added the sentence “The Actigraph was worn continuously throughout the complete recording period.” (Line 128 – 129) in the Recording of motor activity part. We have furthermore improved the reporting on the actigraphy recordings. When reporting the average day’s actigraph recordings, we have supplemented the mean ± SD estimate with range (min – max). We have also updated the footnotes of tab. 1 and 4 to read, “1Number of 24-h sequences analyzed” for the Days1 variable, as well as reported the Days variable in table 3 and 6

#5. It would not be easy to perform validation with such a small number of samples. Validation process seems to require more specific and easy to understand technology and methodology. In particular, in this case, if you repeat the learning and validation several times, the samples will eventually overlap and it may not affect the internal connectivity, which may have an impact on consequences. You need some explanation to overcome this.

A: It is true that the number of samples is small. To this extent, we employed a leave-one-out validation approach to make the most efficient use of the available data. This strategy also provides a better generalization estimate since it assumes that the model does not know anything about the target user, which is a common scenario in real life situations when one wants to use a system out-of-the-box without going into a calibration process.

#6. Please revise the figures and tables to make them readable. It is difficult to recognize.

A: Figures 1 and 2 are updated, but all tables are in line with PLOS ONEs requirements for tables, and according to instructions: “In Word, tables that run off of the manuscript page can be seen using Draft View.”

#7. In the previous studies, authors mentioned that there was no significant difference between unipolar and bipolar depression, but I still have questions about it. Of course, since unipolar depression can be diagnosed as bipolar disorder in the future, it is not easy to make a judgment based on the current diagnosis. However, it is prudent to gather and analyze heterogeneous groups into one group. Analyzing depend on the level of activity, sampling may have its own bias. It is suggested to analyze by dividing unipolar and bipolar. And, if there is data on the normal (euthymic) mood of patients with mood disorders, not normal control people, it is necessary to compare and analyze it. It is also important to distinguish the depressed state of mood disorder from normal people, but it is more important to distinguish the euthymic and depressed mood states of mood disorders.

A: We totally agree with the reviewers’ comments on doing intra-individual comparisons of mood states, but we have yet not had access to such datasets. There have been identified differences between unipolar and bipolar patients when comparing shorter time series (especially in morning periods), but not in longer 24-h series, as stated in the manuscript. We agree that combining unipolar and bipolar depressions is a limitation of our analyzes, but the sample is too small to analyze by dividing the unipolar and bipolar patients. 

#8. Statistics Line 188~

How did you deal with the section that could be thought of as sleep? Did you ever think about an activity level of zero throughout the day regardless of sleep or not? 

A: As we have analyzed 24-h time series, we believe the proportion of Zero activity reflects both sleep patterns and active and non-active periods. Furthermore, the proportion of Zero probably mirrors reduced daytime motor-activity and less complexity in activity patterns typically associated with (retarded) depression. 

#9. Outcome Metrics 197~

You mean a depressed condition? It would be better to describe it a little more clearly. Calling a condition group is easily confusing. It is recommended to describe it a depressed mood.

A: Thank you for identifying this unclear statement, the description is changed according to the recommendation. (Line 211)

#10. Table 1

Even in a healthy control group, it is basically necessary to present and compare the same psychometric values. In this study, MADRS was presented, but the results of this study do not know the mood state of the normal control group.

A: We agree with the reviewer but assessment of MADRS was not done in the control group. This is a limitation that was pointed out in the manuscript (line 493 – 494 in the revised manuscript): “Anyhow, little is known about the control group beyond age, gender and the absence of a history of either affective or psychotic disorders.”

#11. When data related to sleep are analyzed together, some limitations of activity data can be improved. Consideration should be given to analyzing and presenting sleep data.

A: Depression is commonly associated with sleep disturbance, where some depressed people experience increased need for sleep, and others reduced sleep quality and less sleep duration. We chose to analyze a full 24-h time series incorporating sleep patterns. By doing it this way, we have followed the analytic recommendations from a systematic review on activation (Scott J., et al. Activation in bipolar disorders: A systematic review. JAMA Psychiatry. 2017). 

#12. I recommend that you try to train activity data by making it more diverse secondary variables. The strength of this data is that it is a time series. In the introduction, the characteristics of time series data and the necessity of proper analysis were explained, but in the present, only a several MLs were applied. Whether it is an analysis according to the circadian rhythm, the difference between weekdays and weekends, the difference between morning and afternoon, the difference between the most active and non-active periods, the irregularity of activities, etc. I think it is necessary. The author should considers the characteristics of the time series as much as possible, and analyzes according to the circadian rhythm, the difference between weekdays and weekends, the difference between the morning and the afternoon, the difference between the most active and non-active periods, irregularities in activities, etc. You need to do a sophisticated analysis with the possibility of creation.

 A: Regarding utilizing secondary variables, the motivation behind applying CNN analyzing an image representation of the data, was to preserve all the information lost by the feature-extraction. Indeed, the image representation preserves all time series information. We added the following paragraph to the manuscript (line 185 – 189): “The motivation of using a CNN is because with this approach, more information is preserved as compared to a feature-based approach. With the CNN and the chosen representation, the granularity is at the minute level and there is no need to do feature extraction. On the other hand, for the non-CNN based methods, the features were computed on a daily basis which can lead to some information being lost”.

Furthermore, we have analyzed 24-h time series expressing a full circadian cycle, and do think diurnal variations of active and non-active periods and sleep patterns are incorporated. Although we principally agree with a need for addressing difference between weekdays and weekends in such data, we did not find this relevant for our sample, as our healthy controls are mainly shifts working hospital ward employees, working daytime, evenings and weekends, and we have no information about their work-schemes. The healthy control group is described in the previous papers on this sample. We have now added the sentence “The majority were shift working hospital staffs” at line 119. 

The comprehensive analytical approach that the reviewer desires would require a self-reported systematic description of each individual's social rhythms. We agree that this would be a very expedient and interesting experiment, but should probably be done using intra-individual comparisons of various mood states. We have supplemented the manuscript with the following text (line 519 - 523). “ML analyzes in such an intra-individualistic sample, may provide analyzes beyond the 24-h circadian cycles studied in this experience. For instance, differences between morning and evening, active and non-active periods and sleep patterns, as well as activity differences between weekdays and weekends, could be more feasible in intra-individualistic samples”. Placed at the end of the paragraph discussing the need for intra-individualistic data on depression.

At the same time, we agree that additional features may provide more information to the predictive models. As suggested by the reviewer, it would be interesting to try additional data-driven approaches and feature engineering to extract more features related to weekdays, weekends, day and night patterns, etc. and perform a feature importance analysis to detect what are the most relevant variables when discriminating between classes. These types of analyses have been proposed as a future direction either by us or by other researchers since the dataset is public. We have added the following text in the discussion section (line 523 – 526): “In the present work, we have furthermore analyzed a possible set of statistical features extracted from activity time series, and displayed how accurate machine learning models can be trained with those. As a future direction, the utilized public dataset provides the possibility to explore the use of additional statistical features.”

---

## [Decision Letter · Decision Letter 1]

11 Aug 2020

Applying machine learning in motor activity time series of depressed bipolar and unipolar patients compared to healthy controls.

PONE-D-20-09443R1

Dear Dr. Jakobsen,

We’re pleased to inform you that your manuscript has been judged scientifically suitable for publication and will be formally accepted for publication once it meets all outstanding technical requirements.

Kind regards,

Kyoung-Sae Na, M.D.

Academic Editor

PLOS ONE

Additional Editor Comments (optional):

Reviewers' comments:

Reviewer's Responses to Questions

**Comments to the Author**

1. If the authors have adequately addressed your comments raised in a previous round of review and you feel that this manuscript is now acceptable for publication, you may indicate that here to bypass the “Comments to the Author” section, enter your conflict of interest statement in the “Confidential to Editor” section, and submit your "Accept" recommendation.

Reviewer #1: All comments have been addressed

2. Is the manuscript technically sound, and do the data support the conclusions?

Reviewer #1: Yes

3. Has the statistical analysis been performed appropriately and rigorously? 

Reviewer #1: Yes

4. Have the authors made all data underlying the findings in their manuscript fully available?

Reviewer #1: Yes

5. Is the manuscript presented in an intelligible fashion and written in standard English?

Reviewer #1: Yes

6. Review Comments to the Author

Reviewer #1: (No Response)

7. PLOS authors have the option to publish the peer review history of their article (what does this mean?). If published, this will include your full peer review and any attached files.

Reviewer #1: **Yes: **Jonas Busk

---

## [Editor Report · Acceptance letter]

14 Aug 2020

PONE-D-20-09443R1 

Applying machine learning in motor activity time series of depressed bipolar and unipolar patients compared to healthy controls. 

Dear Dr. Jakobsen:

I'm pleased to inform you that your manuscript has been deemed suitable for publication in PLOS ONE. Congratulations! Your manuscript is now with our production department. 

Kind regards, 

on behalf of

Dr. Kyoung-Sae Na 

Academic Editor

PLOS ONE